# Physics-informed Hamiltonian learning for large-scale optoelectronic property prediction

Martin Schwade ®[1], Shaoming Zhang[1], Frederik Vonhoff ®[1], Frederico P. Delgado[1] & David A. Egger ®[1,2] ✉

Predicting optoelectronic properties of large-scale atomistic systems under realistic conditions is crucial for rational materials design, yet computationally prohibitive with first-principles simulations. Recent neural network models have shown promise in overcoming these challenges, but typically require large datasets and lack physical interpretability. Physics-inspired approximate models offer greater data efficiency and intuitive understanding, but often sacrifice accuracy and transferability. Here we present HAMSTER, a physics-informed machine learning framework for predicting the quantum-mechanical Hamiltonian of complex chemical systems. Starting from an approximate model encoding essential physical effects, HAMSTER captures the critical influence of dynamic environments on Hamiltonians using only few explicit first-principles calculations. We demonstrate our approach on halide perovskites, achieving accurate prediction of optoelectronic properties across temperature and compositional variations, and scalability to systems containing tens of thousands of atoms. This work highlights the power of physics-informed Hamiltonian learning for accurate and interpretable optoelectronic property prediction in large, complex systems.

Quantum-mechanical predictions of large-scale chemical systems are of key importance to accelerating progress in diverse fields ranging from quantum biology[1] and materials science[2] to drug design[3]. In optoelectronic materials, which are key components in photovoltaic and lightning applications, computations of large-scale systems are essential to capture real-world factors such as the presence of defects and dynamic disorder that strongly influence device behavior through charge transport and recombination processes. Ideally, one would predict the electronic structure of such large and disordered systems at finite temperature directly from first-principles. Achieving this would move quantum-mechanical modeling beyond the prevailing paradigm, which assumes an ideal lattice at 0 K, Bloch-wave electronic states, and perturbative treatments of critical effects such as electron-phonon or electron-defect interactions. Computational capabilities that go beyond such frameworks and accurately capture thermal effects on optoelectronic properties represent a crucial step towards realistic materials modeling under operating conditions, ultimately advancing the design of next-generation optoelectronic technologies.

Progress in machine learning force fields (MLFFs) suggests promising routes to overcome steep limitations in the modeling of large-scale molecules and materials. In particular, MLFFs enable first-principles accuracy of molecular dynamics (MD) simulations at dramatically reduced computational cost[4–14]. While MLFFs can accurately predict atomic trajectories at specified thermodynamic conditions, they typically lack the ability to provide insights into electronic structure, an increasingly important demand given the recent availability of highly accurate MD trajectories of large-scale quantum systems. Relying on explicit first-principles electronic-structure theory, such as density functional theory (DFT), remains impractical because of its unfavorable scaling of computational cost with increasing system size.

[1]Physics Department, TUM School of Natural Sciences, Technical University of Munich, Garching, Germany. [2]Atomistic Modeling Center, Munich Data Science Institute, Technical University of Munich, Garching, Germany. ✉e-mail: david.egger@tum.de

Recent work[15,16] suggests that equivariant machine learning (ML) provides a powerful framework to sidestep explicit first-principles calculations of large-scale chemical systems by learning and predicting the quantum-mechanical Hamiltonian. Such Hamiltonian learning frameworks require training an ML model with first-principles data, from which one can then predict the Hamiltonian matrix using atomic coordinates as input to the model. Omitting explicit first-principles calculations, one can then, in principle, calculate many quantum-mechanical observables of large-scale chemical systems.

Relevant in this context, learning the Hamiltonian is challenging because it depends not only on a system's atomic structure but also on the chosen representation, so there is no single unique mapping from structure to matrix elements. However, equivariant ML models respecting fundamental physical transformations, such as rotations and translations, ensure that predictions transform consistently under symmetry operations, and thereby significantly improve data efficiency and generalization. Specifically, equivariant neural networks (ENNs) were recently developed as particularly promising frameworks for Hamiltonian learning[17–19]. Although ENNs improve data efficiency by encoding symmetry and therefore require fewer samples than comparable non-equivariant models, they may still demand substantial amounts of training data to capture complex dynamical behavior, such as that exhibited in multi-component materials at finite temperature. Related to this limitation, existing ENNs for Hamiltonian learning usually rely on access to the real-space Hamiltonian represented in a consistent basis, which restricts their applicability to specific DFT frameworks requiring careful benchmarking[20].

To alleviate some of these challenges, Gu et al. recently combined a tight-binding (TB) model with deep learning to predict Hamiltonians, demonstrating accurate electronic-property predictions for large-scale simulations using DFT eigenvalues as training data[21]. Although this suggests that incorporating approximate physical models into the ML workflow can substantially improve prediction accuracy, there is still little understanding of how to best integrate physics-aware approaches with emerging ML architectures or hybrid multiscale simulations for more complex material classes. Thus, there is a need for new approaches seeking to combine the efficiency of physics-based approximations with the flexibility of ML while maintaining transferability across material classes.

In this work, we address these critical limitations by developing a physics-informed ML framework for predicting the quantum-mechanical Hamiltonian of large-scale chemical systems. Our approach starts from considering the traditional TB model, which is physically motivated and requires little data for fitting, yet falls short in predictive accuracy and transferability. We augment the physical model with an ML scheme that captures the key effect of the dynamic environment on the electronic Hamiltonian. We find that our integrated model, HAMSTER (**H**amiltonian-learning **A**pproach for **M**ultiscale **S**imulations using a **T**ransferable and **E**fficient **R**epresentation), achieves first-principles accuracy with only a small number of explicit first-principles calculations, offering substantially greater data efficiency than existing frameworks. We assess the performance of HAMSTER on halide perovskite semiconductors, a challenging case due to their soft lattices, dynamic disorder, and chemical complexity, well beyond the systems typically considered in prior work. We show that the model yields accurate electronic-property predictions, consistent with first-principles and experimental results, across diverse compositions and temperatures for large-scale systems containing tens of thousands of atoms.

## Results
### Physics-informed Hamiltonian-learning model
The objective of Hamiltonian learning is to predict the Hamiltonian matrix using the atomic coordinates as input. For chemical systems containing many atoms, the Hamiltonian is a large matrix that encodes rich quantum-mechanical interactions. Predicting it from scratch with ML is a challenging task that requires fitting many parameters, which is computationally expensive. Our goal is to enhance the data efficiency of Hamiltonian learning by incorporating underlying physical principles into the model prior to the ML procedure. To achieve it, we suggest a ΔML approach, where the ML model learns differences between results from a physical model and the ground truth instead of initializing the fitting procedure from scratch (see Fig. 1a).

In our HAMSTER approach, we therefore start from the well-known TB model that requires a small amount of first-principles data as input and provides an approximate yet physically motivated description of the Hamiltonian. Specifically, assuming periodic boundary conditions, we write the matrix elements of the TB Hamiltonian in $k$-space as

$$H_{ij}^{\mathbf{k}} = \sum_{\mathbf{R}} e^{i\mathbf{k}\cdot\mathbf{R}} \underbrace{\int d^3 r \chi_i(\mathbf{r})\hat{H}_{\text{eff}}\chi_j(\mathbf{r}-\mathbf{R})}_{:= H_{ij}^{\mathbf{R}}}, \tag{1}$$

where $\chi_i$ is the $i$-th electron orbital, $\mathbf{R}$ a real-space lattice vector, and $\hat{H}_{\text{eff}}$ an effective independent-particle Hamiltonian. The indices $i$ and $j$ are composite indices that specify both the orbital and the atomic site. To identify different types of interactions, it is useful to write the effective Hamiltonian as

$$\hat{H}_{\text{eff}} = \frac{p^2}{2m} + \sum_{k=1}^{N_{\text{ion}}} V_k(\mathbf{r}-\mathbf{r}_k) + H_{\text{soc}}, \tag{2}$$

where $H_{\text{soc}}$ encapsulates spin-orbit coupling (SOC) effects (see "Methods" section), and the potential is expressed in terms of contributions by atoms located at sites $k$[22].

Using Eq. (2) in Eq. (1), we can identify different types of matrix elements: (i) $i = j$ corresponds to on-site matrix elements; (ii) $i = j \neq k$ to environment effects for on-site elements; (iii) $i \neq j = k$ or $i = k \neq j$ corresponds to off-site matrix elements; (iv) $i \neq j \neq k$ to environment effects for off-site elements; and (v) SOC matrix elements. Contributions due to (ii) and (iv) are often neglected in common TB models, which is known as the two-center approximation[23–25]. Using this approximation, in our TB model[26], we compute the two-center contributions as

$$H_{ij}^{\mathbf{R}} = \sum_{v} C_{vij}^{\mathbf{R}} V_v(\Delta r) V_v^0, \tag{3}$$

where $C_{vij}^{\mathbf{R}}$ takes into account the relative orientation in the $v$-th orbital interaction, $V_v(\Delta r)$ accounts for the distance between the orbitals, and $V_v^0$ are adjustable parameters, hereafter referred to as TB parameters.

At finite temperatures, atoms in molecules and materials are constantly in motion, which leads to fluctuations in their local environments. Starting from our TB model, we therefore augment the Hamiltonian in a ΔML step[27] (see Fig. 1a) to capture the effects of each atom's dynamic environment via interaction terms (ii) and (iv). To this end, each non-zero matrix element in the Hamiltonian of Eq. (3), determined by a cutoff radius $r_{\text{cut}}$, is associated with a descriptor vector, $\mathbf{x}_{ij}^{\mathbf{R}}$, whose specific construction is detailed below. From the full set of descriptor vectors, we select a subset of size $N_p$ using the k-means clustering algorithm. These serve as kernel support points and are used to predict the corrections for remaining or unseen matrix elements (see "Methods" for details on the sampling procedure).

We employ a radial basis function kernel model, where each kernel support point, $x_n$, is associated with a learnable parameter, $h_n$. Given a trial environment descriptor vector, $\mathbf{x}_{ij}^{\mathbf{R}}$, the ML correction for the respective matrix element in Eq. (3) is calculated as

$$\delta H_{ij}^{\mathbf{R}} = \sum_{n=1}^{N_p} h_n \exp\left(-\frac{\left|\mathbf{x}_n - \mathbf{x}_{ij}^{\mathbf{R}}\right|^2}{2\sigma^2}\right), \tag{4}$$

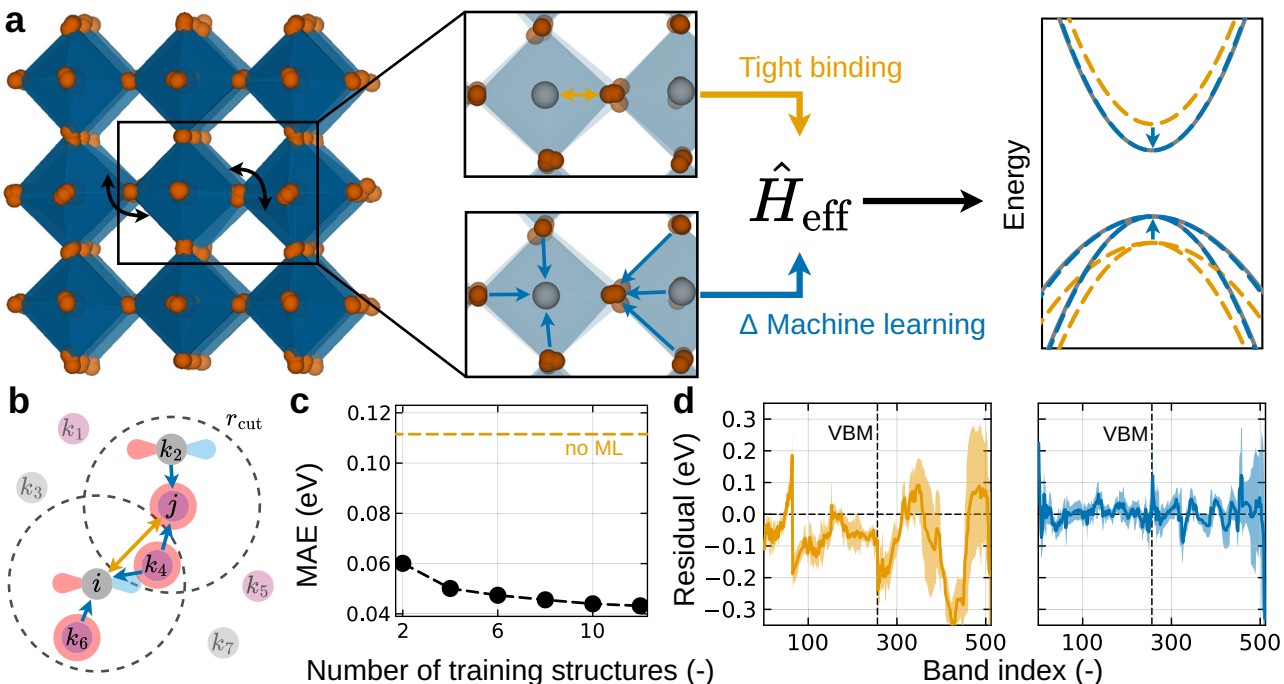

**Fig. 1 | Overview of the physics-informed Hamiltonian learning model and workflow with results for GaAs. a** Illustration of pairwise and environment-dependent electronic interactions arising from structural fluctuations in a chemical system and their effect on the electronic structure captured via an effective Hamiltonian, $\widehat{H}_{\text{eff}}$, in the HAMSTER approach. It combines a physical model and environment-dependent machine learning terms. **b** Schematic visualization of the environment descriptor for the matrix element between atoms $i$ and $j$, which treats local environments of the two atoms separately. Different atomic species are indicated by purple and gray colors. Atoms within a distance $r_{\text{cut}}$, which is indicated by dashed circles, are labeled as $k_x$, with $x = 1, 2, 3, \ldots$ The $s$ and $p$ orbitals of selected atoms are shown schematically as red circles and red-blue handles, respectively. **c** Validation error, calculated as mean absolute error (MAE) across all eigenvalues, for a HAMSTER model trained on an increasing number of training structures for GaAs at 400 K. **d** Comparison of residuals, defined as the difference between energy eigenvalues computed with the respective model and density functional theory (DFT). Results are shown for a pristine tight-binding (orange) and the HAMSTER model (blue), averaged over $k$-points and 100 snapshots, at 400 K. Dashed vertical lines indicate the valence band maximum (VBM).

where the kernel width, $\sigma$, acts as a hyperparameter that has to be specified before optimization, as is the case with $N_p$. With $\widehat{H}_{\text{eff}}$ defined, one may proceed with kernel ridge regression, using first-principles Hamiltonian matrix elements as ground-truth values. Instead, we propose using energy eigenvalues obtained from first-principles calculations as training targets and optimize the model parameters via gradient descent (see "Methods" section for more details).

A key remaining part of the model is selecting descriptor vectors for Hamiltonian matrix elements. Various descriptors were proposed in prior studies[16,28–30]. A significant advantage is the choice of a physical model for the Hamiltonian, which already accounts for changes in the distance and relative orientation between orbitals $i$ and $j$ (see Eq. (3)). Then, the ML correction only needs to predict finer dynamic modifications of matrix elements via Eq. (4). To be physically meaningful, the descriptors should vary smoothly with changes in atomic positions and respect the symmetries of the Hamiltonian. Accordingly, we adopt a relatively simple descriptor that encodes both the interaction type and surrounding atomic environment (see Fig. 1b): we approximate the environment of $H_{ij}$ as a combination of local environments around atoms $i$ and $j$, treated independently. The local environment of each atom $i$ is computed as

$$h_{\text{env}, i} = \sum_{k \neq i} f_{\text{cut}}(|\mathbf{r}_i - \mathbf{r}_k|) \int \mathrm{d}^3 r \, \chi_i(\mathbf{r} - \mathbf{r}_i) \chi_k(\mathbf{r} - \mathbf{r}_k), \quad (5)$$

where the sum runs over all atoms and orbitals, indexed by $k$, within a cutoff radius, $r_{\text{cut}}$. The smooth cutoff function, $f_{\text{cut}}(r) = \frac{1}{2}[\cos(\frac{\pi r}{r_{\text{cut}}}) + 1]$, ensures a gradual decay of contributions from distant atoms. The local environment is identical for all lattice translation vectors, $\mathbf{R}$, as a consequence of translational symmetry. Note that the integrals in Eq.

(5) reduce to those in Eq. (3) when $V_\nu = 1$, and therefore need to be evaluated only once for both the TB and ML contributions.

To provide the model with information about the interaction type, we augment the descriptor vector with several structural features: the equilibrium distance between atoms $i$ and $j$, the angular orientation of their respective orbitals relative to the connecting vector, $\mathbf{r}_{ij}$, and the atomic numbers $Z_i$ and $Z_j$. These quantities can either be kept fixed during atomic motion or updated dynamically, thereby providing the model with the flexibility to accommodate additional physical effects such as thermal lattice expansion. This design ensures that the ML model focuses on learning changes in the local environment, thereby improving data efficiency by reducing the complexity of the learning task. The descriptor vector then reads

$$x_{ij}^{\mathbf{R}} = \left( Z_i, Z_j, \Delta r_{ij}^{\mathbf{R}}, \theta_i^{\mathbf{R}}, \theta_j^{\mathbf{R}}, \varphi_{ij}^{\mathbf{R}}, h_{\text{env}, i}, h_{\text{env}, j} \right)^{\mathsf{T}}. \quad (6)$$

Here, $\Delta r_{ij}^{\mathbf{R}}$ is the distance between atoms $i$ and $j$ with the translation vector $\mathbf{R}$ applied to atom $j$, $\theta_x^{\mathbf{R}}$ ($x = i, j$) is the angle between the orientation of the orbital on atom $x$ and the bonding axis, and $\varphi_{ij}^{\mathbf{R}}$ is the angle between the orbital orientations on atoms $i$ and $j$.

Finally, we emphasize that the ordering of values within the descriptor vector in Eq. (6) is critical, as inconsistent arrangements can violate the symmetry properties of the Hamiltonian. To preserve these symmetries, the descriptor is constructed such that its structure remains invariant under operations that should produce equal matrix elements, for example, the relation $H_{ij}^{\mathbf{R}} = H_{ji}^{-\mathbf{R}}$ (see, e.g., ref. 16 for a detailed discussion of Hamiltonian symmetry properties). This should be distinguished from equivariance, which instead ensures that the

model output transforms consistently under symmetry operations, as employed in equivariant models[13–15,18,31].

We apply the HAMSTER approach to bulk GaAs as a test case, starting from the above-described TB model (see ref. 26 for full details). Figure 1c shows that training the ΔML method requires only a few explicit first-principles calculations of MD-generated structures at a temperature of 400 K for the validation error, calculated as mean absolute error across the eigenvalues (MAE, see "Methods" section), to converge. Figure 1d shows that systematic deviations in the TB eigenvalue spectrum at 400 K are markedly reduced when adding the ΔML correction, yielding sub-50 meV accuracy for many of the eigenvalues in comparison to DFT. These results suggest that the primary reason for the inaccuracies of the TB model stems from its omission of dynamic environment effects, which are effectively addressed through the addition of the ML correction in a data-efficient way.

## Physics-informed Hamiltonian learning for halide perovskites

Having established the effectiveness of the HAMSTER approach for a relatively simple test case, we now turn to the significantly more complex halide perovskite semiconductors. These materials have attracted significant attention due to their promising optoelectronic properties and applications in photovoltaics and light-emitting devices[32,33]. At the same time, they exhibit complex vibrational properties, characterized by anharmonic, overdamped phonons and unusual electron-phonon coupling[34–37], which are generally challenging to capture in atomistic models[38,39]. Furthermore, the halide $p$-orbitals in the $ABX_3$ perovskite crystal structure are embedded in two distinctly different environments, as these orbitals are oriented either along a bonding axis or perpendicular to it. These aspects render the computational modeling of perovskite optoelectronic properties at finite temperature particularly challenging. We consider cesium lead bromide ($CsPbBr_3$) in its cubic phase as a prototypical inorganic halide perovskite material and investigate the performance of HAMSTER.

We first construct a TB model for $CsPbBr_3$ as a physically meaningful approximation of the Hamiltonian. The model includes bromine $p$-orbitals, lead $s$- and $p$-orbitals, and cesium $s$-orbitals in the basis, as these dominantly contribute to the electronic structure close to the valence band maximum (VBM) and conduction band minimum (CBM). Furthermore, we explicitly account for SOC in the TB Hamiltonian, given that it plays a crucial role in the electronic structure of halide perovskites. Further details regarding the model and its optimization are described in the "Methods" section.

To determine an appropriate training set size for ΔML, we train our model on an increasing number of structures drawn from an MD trajectory at a temperature of 425 K. Sampling an equal number of kernel support points from each structure, we compare the use of $r_{cut}$ = 5.5 Å and $r_{cut}$ = 6.2 Å (see Eq. (5)) in two separate optimizations to assess the impact of the interaction range on model performance.

The learning curves (see Fig. 2a) show that the validation error plateaus after using around 10 training structures, reaching an accuracy for the eigenvalues that drops below 50 meV. While this level of accuracy is comparable to that reported in a recent study[21] using a neural-network scheme for elemental and binary semiconductor compounds, we emphasize that our approach needs far fewer training structures for the considerably more complex halide perovskite $CsPbBr_3$, well below the few hundred typically required by neural-network-based Hamiltonian learning models[17,21,31]. Remarkably, for this material, HAMSTER still generalizes reasonably well with as few as two training structures, remaining within 0.03 eV of the training error, highlighting the markedly superior data efficiency of our approach. The mild overfitting at very low training set sizes, notable through the small gap between training and validation error, disappears as the training set increases.

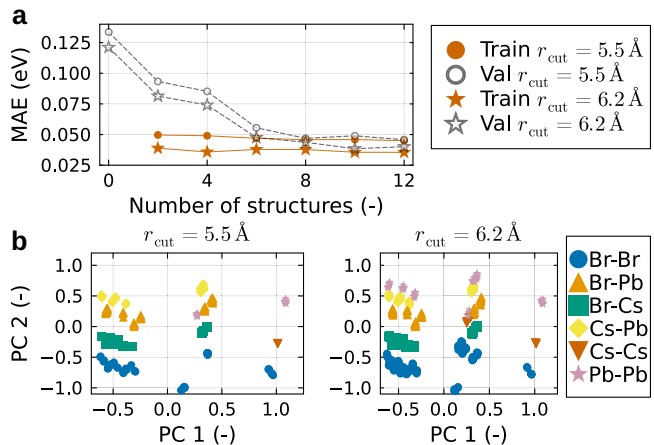

**Fig. 2 | Training and descriptor analysis for CsPbBr₃. a** Dependence of training (dark orange) and validation (gray) loss on the number of training structures. The loss is computed as the mean absolute error (MAE) between density functional theory (DFT) and HAMSTER energy eigenvalues. The data point at zero training structures corresponds to the underlying tight-binding model without machine-learning corrections. **b** Principal component analysis of the kernel support points for 12 training structures. In both panels, we compare two different cutoff radii ($r_{cut}$) for the interaction range, namely 5.5 Å and 6.2 Å.

Furthermore, we observe that the prediction errors decrease with increasing cutoff radius (see Fig. 2a). This is expected, as a larger cutoff captures more interactions, increasing the expressiveness of the model by introducing additional non-zero Hamiltonian elements. However, this improvement in accuracy comes at the cost of increased computational effort, as the Hamiltonian becomes denser as $r_{cut}$ increases. Thus, selecting the cutoff radius requires balancing accuracy against computational efficiency.

We analyze the structure of the descriptor space to gain insight into the model's increased expressiveness when a larger $r_{cut}$ is used. To this end, we perform a principal component analysis (PCA, see "Methods") on the sampled descriptors for both cutoff radii, which projects the eight-dimensional descriptor vectors onto a two-dimensional subspace while preserving as much of their variance as possible. This enables a clear visualization of their distribution (see Fig. 2b). In this representation, descriptor vectors that encode types of interactions lie close together. Since each descriptor component corresponds to a specific orbital interaction in the Hamiltonian (see Eq. (6)), the PCA projection groups and separates regions of descriptor space according to the orbital couplings they represent.

For each interaction type, we observe multiple distinct clusters, which can be associated with specific, physically-interpretable orbital coupling types, such as sp$\sigma$ or pp$\pi$. When increasing the cutoff radius, new clusters emerge that correspond to additional interactions between atoms in neighboring unit cells, including Pb-Pb, Cs-Cs, and Br-Br couplings. This not only demonstrates the ability of the ML model to distinguish between interaction types but also its capability to learn physically-interpretable corrections according to additional interactions, which further enhances accuracy.

## Transferability across temperatures and system sizes

Having trained and benchmarked HAMSTER at its reference temperature, we now examine its transferability across temperatures to assess robustness at varying thermal conditions. Thermal variations of optoelectronic properties in halide perovskites are notoriously difficult to capture computationally. A prime example is the band gap, whose anomalously weak temperature dependence defies the trends seen in conventional semiconductors and poses a significant challenge for predictive modeling[40–42].

Before turning to large-scale simulations, we apply the model to MD trajectories of a $2 \times 2 \times 2$ supercell of $CsPbBr_3$ for benchmarking against DFT data, using $r_{cut} = 5.5$ Å in our model. Figure 3a compares the temperature dependence of the band gap in $CsPbBr_3$ as obtained with HAMSTER and DFT. Although the model was trained solely on structures from an MD trajectory at 425 K, its predictions remain in close agreement with DFT also at higher temperatures. Across all three considered temperatures, the difference between the DFT-computed band gap and the value predicted by HAMSTER remains below 50 meV, and the MAEs of energy eigenvalues are 0.047 eV (425 K), 0.049 eV (525 K), and 0.057 eV (625 K). Furthermore, we find that the fluctuations of instantaneous band gaps, gauged by the standard deviation at each temperature, are similar in DFT and HAMSTER. These results underscore the transferability of our model over a considerable temperature range within the cubic phase of $CsPbBr_3$.

Figure 3b presents the average residual for each band obtained in the pristine TB model (left panel) in comparison to HAMSTER (right panel) at a temperature of 425 K. Without accounting for the effect of the dynamic environment, the energy of certain bands appears systematically over- or underestimated, indicating inherent limitations of the TB model in capturing band-specific dynamic fluctuations in the electronic structure. Incorporating the environment-dependent interactions via the ΔML step, these systematic deviations are significantly reduced, with residuals now appearing much less amplified and more randomly distributed around zero. A notable exception remains for the uppermost conduction bands, where a consistent mismatch persists. This can be attributed to their reduced weighting during optimization (see "Methods" section). Overall, these results demonstrate the strength of the physics-informed Hamiltonian learning framework, which combines the ability of TB models to capture the essential features of the electronic structure with the flexibility of ML to incorporate subtle but crucial environmental and multi-center effects.

A key advantage of our approach is that it enables simulations of system sizes far beyond the practical reach of explicit first-principles methods such as DFT. Modeling of larger systems can reveal distinct dynamical behavior and, by reducing finite-size effects, provide a more realistic description of the material. To explore this regime, we train an MLFF for $CsPbBr_3$ (see "Methods") and perform MD simulations for a $16 \times 16 \times 16$ supercell of $CsPbBr_3$ consisting of 20,480 atoms. Using the HAMSTER model, we subsequently predict Hamiltonians and compute band gaps by extracting eigenvalues at the VBM and CBM via the Lanczos algorithm as implemented in ARPACK[43]. Figure 3a shows that the mean value of the band gap at 425 K closely matches the one obtained for the smaller supercell. At the same time, the sizable band-gap fluctuations seen for the smaller supercells are almost entirely

averaged out, yielding a much narrower distribution and a more realistic description of the material at finite temperature. This improved statistical convergence allows us to assess the temperature dependence of the band gap. While the absolute value of the band gap is affected by the known inaccuracies of PBE-level DFT, the resulting temperature dependence can still be meaningfully compared to experiment. For the $16 \times 16 \times 16$ supercell, we obtain a slope of $3.95 \times 10^{-4}$ eV K$^{-1}$, in good agreement with the experimental value of $3.41 \times 10^{-4}$ eV K$^{-1}$ reported in ref. 44. We note, however, that thermal expansion is not included in the simulation and that the temperature range considered here extends beyond that of the experimental study, both of which are expected to affect the band gap values.

## Application to additional compositions

To demonstrate the transferability of our workflow to other perovskites, we apply the HAMSTER model to $MAPbBr_3$. This choice is motivated by a recent study showing that in the related perovskite $FAPbI_3$, the supercell size was particularly important for reaching accurate band gap predictions, primarily due to molecular asymmetry, requiring simulation cells with at least 6000 atoms[45].

We optimize the model parameters following the same procedure as for $CsPbBr_3$ (see "Methods" section), again using $r_{cut} = 5.5$ Å. However, for this material, we do not include orbitals stemming from the cation and perform classical force-field MD calculations (see "Methods" section). When applied to an independent test set consisting of 100 MD snapshots of a $2 \times 2 \times 2$ supercell at 300 K, the model reaches an MAE of 0.060 eV relative to DFT. This demonstrates that the model maintains a level of accuracy comparable to that achieved for $CsPbBr_3$.

Figure 4a compares the band gaps of $MAPbBr_3$ obtained from DFT and HAMSTER across different supercell sizes at 300 K. Leveraging the computational efficiency of HAMSTER, we extend our simulations to $16 \times 16 \times 16$ supercells comprising almost 50,000 atoms. The predicted band gap of $MAPbBr_3$ decreases by approximately 0.15 eV as the supercell size increases, in line with a recent report on $FAPbI_3$[45]. Agreement between HAMSTER and DFT is observed up to $4 \times 4 \times 4$ supercells (768 atoms), beyond which direct DFT calculations become impractical.

Figure 4 presents the variation of the band gap with temperature as predicted by HAMSTER across different supercell sizes, together with experimental data for comparison. The objective here is not to reproduce the experimental trend with exact quantitative accuracy, but to employ it as a benchmark for assessing the reliability of the model. Although minor deviations between results obtained with HAMSTER and experimental data remain, the predicted temperature dependence captures the experimental behavior well, with the agreement becoming more favorable at larger supercell sizes. This

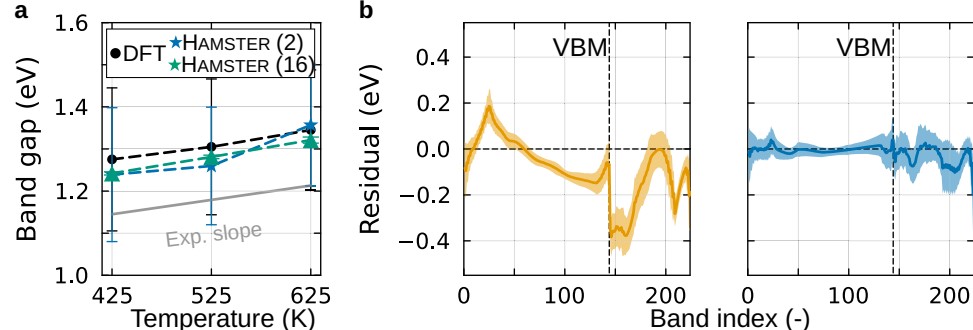

**Fig. 3 | Transferability of the model across temperatures and large-scale calculations. a** Band gap of $CsPbBr_3$ computed with density functional theory (DFT, black) and HAMSTER (blue) for a $2 \times 2 \times 2$ supercell of $CsPbBr_3$. Results obtained using HAMSTER for a $16 \times 16 \times 16$ supercell are shown as well (green). Error bars indicate the standard deviations across molecular dynamics snapshots. The gray line represents the slope of the fitting function derived from experimental data[44]. **b** Comparison of residuals relative to DFT data for $CsPbBr_3$ at 425 K, averaged over 100 snapshots and all $k$-points. The pristine tight-binding (TB, orange) and the HAMSTER model (blue) are shown, with shading representing the standard deviation.

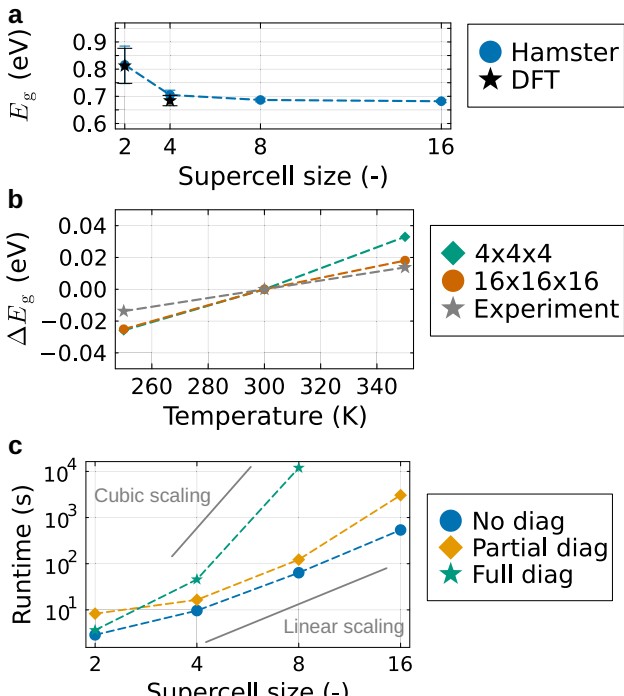

**Fig. 4 | Band gap of MAPbBr$_3$ across supercell sizes and temperatures with computational scaling. a** Band gap computed with HAMSTER (blue) and density functional theory (DFT, black) for varying supercell sizes at a temperature of 300 K. Error bars show the standard deviation across molecular dynamics snapshots. The $x$-axis indicates the supercell dimension $n$, corresponding to an $n \times n \times n$ replication of the cubic cell. **b** Temperature-induced change in band gap, $\Delta E_g$, references to the band-gap value at a temperature of 300 K, computed for varying supercell sizes (green: $4 \times 4 \times 4$; dark orange: $16 \times 16 \times 16$) with HAMSTER and compared to an experimentally-determined fit function (gray)[44]. **c** Runtime of HAMSTER evaluated on 10 structures as a function of cell size. Timings are shown for full diagonalization of the Hamiltonian (full diag; green stars), partial diagonalization of six eigenvalues around the valence band maximum using the Lanczos algorithm (partial diag; orange diamonds), and construction and saving of the Hamiltonian without diagonalization (no diag; blue circles). A constant Julia compilation time of 44 s has been subtracted. Reference lines indicating linear and cubic scaling are shown in gray.

consistency between experiment and computation supports the robustness and overall validity of the approach.

To assess the computational performance of HAMSTER, we benchmark the total runtime for three calculation modes: (i) full diagonalization of the Hamiltonian, (ii) Lanczos-based partial diagonalization targeting six eigenvalues near the VBM, and (iii) construction and output of the Hamiltonian matrix in Eq. (1) without solving the eigenvalue problem. For each benchmark, the model is evaluated for 10 independent structures, using the same configurations as in Fig. 4, and all calculations are performed on a single compute node using two Intel Xeon Platinum 8168 CPUs. Full diagonalization of the largest system size ($16 \times 16 \times 16$) is not tractable on the chosen, rather modest hardware and is therefore omitted. The resulting runtimes are shown in Fig. 4c. Since the benchmarks include the first invocation of each function, we fit a function, $T(x) = T_0 + ax^n$, to the timings and subtract the constant time $T_0 = 44$ s, i.e., the initial Julia compilation overhead, which disproportionately affects benchmarks with short execution times.

The observed scaling behavior is consistent with theoretical expectations. The construction of the Hamiltonian exhibits nearly ideal linear scaling with system size. Partial diagonalization remains significantly faster than full diagonalization, but deviates from linear behavior, showing an effective scaling exponent of approximately 1.8. This deviation originates from the shift-invert transformation, which can reduce the sparsity of the Hamiltonian. In contrast, full diagonalization yields a scaling exponent of 2.8, close to the cubic scaling expected for conventional DFT implementations. It is worth noting that the absolute computational cost of DFT calculations is substantially higher, primarily due to the overhead associated with repeated self-consistent field iterations and the much larger underlying basis sets. Additional speedups for diagonalization could be achieved through more advanced algorithms, highly parallel implementations, or GPU-accelerated solvers.

Within our approach, the wall time for the largest system considered remains below one hour, demonstrating that HAMSTER can efficiently handle large-scale systems even on modest hardware. When only the Hamiltonian is required, and eigenvalues are not computed, genuinely linear scaling can be achieved. Moreover, many important physical quantities, such as transport properties or the density of states, can be evaluated with linear-scaling methods that exploit the sparsity of the Hamiltonian[46,47]. Combining our Hamiltonian learning framework with such techniques, therefore, provides a promising route toward efficient simulations of large-scale systems.

## Discussion

This work demonstrates that physics-informed ML can address the challenge of predicting optoelectronic properties in realistic, large-scale atomistic systems. By introducing HAMSTER, a Hamiltonian-learning framework that combines approximate physical models with ML and uses a small amount of first-principles input, we achieved accurate, interpretable, and scalable predictions across temperatures, compositions, and system sizes of up to tens of thousands of atoms. These results highlight the power of embedding physical knowledge into Hamiltonian-learning models to enable quantitative materials predictions under realistic conditions.

Compared with purely data-driven neural networks for Hamiltonian learning, which often require extensive training sets and provide limited physical transparency, we demonstrated that HAMSTER achieves comparable accuracy with only modest first-principles input while maintaining an interpretable Hamiltonian representation. In contrast to conventional approximate models, our approach retains accuracy across temperatures, compositions, and system sizes. This balance of efficiency, accuracy, and interpretability positions physics-informed Hamiltonian learning as a practical strategy for investigating complex materials that remain inaccessible to direct first-principles methods.

The class of halide perovskite materials studied in this work provides a stringent test case, as they are chemically and structurally complex materials whose optoelectronic properties are strongly influenced by dynamic disorder and thermal fluctuations. The success of HAMSTER in this setting suggests that the framework is broadly applicable to other materials where such effects play a decisive role. Its combination of a physically meaningful, approximate model and an ML framework capturing subtle but important dynamic effects in the electronic structure provides a data-efficient pipeline for optoelectronic property predictions. Such is particularly suited to problems that are otherwise impractical for direct first-principles simulation, such as the influence of temperature, compositional disorder, and large system size on functional material properties. By enabling quantitative modeling under realistic conditions, physics-informed Hamiltonian learning opens opportunities for studying phenomena ranging from carrier transport in semiconductors to defect physics in complex crystalline materials.

An important consideration that follows from this capability is chemical transferability. In principle, transfer learning is feasible when the source and target systems share similar local environments and orbital character, allowing pre-trained parameters to serve as

meaningful initializations. This may hold for materials that differ only by modest compositional changes or chemically similar substitutions. However, when electronic structure or orbital symmetries differ substantially, previously learned parameters are unlikely to remain valid, and retraining becomes necessary. At the same time, these considerations highlight the longer-term potential for developing a more general model that spans broader classes of materials and can be efficiently adapted to new systems with minimal additional training.

While in this work we deliberately adopted a simple ML architecture, future extensions could tackle physics-informed frameworks that leverage more powerful global approaches, such as ENNs, to capture non-local electronic effects beyond the present description. Similarly, although our pilot implementation presented in this work relied on semi-local DFT, the modest requirement for first-principles input data makes it feasible to incorporate more advanced electronic-structure methods. These could include hybrid functionals or many-body perturbation theory, potentially extending the framework to include excited-state effects that can play an important role in the modeling of optoelectronic properties. Such enhancements could further enhance prediction accuracy. These directions, together with applications to broader classes of complex materials, highlight the potential of physics-informed Hamiltonian learning as a versatile tool for advancing quantitative materials modeling under increasingly realistic conditions.

## Methods

### Dataset of electronic-structure calculations

All DFT calculations were performed using the VASP code[48] employing the PBE functional[49] and accounting for SOC in the case of the halide perovskites. Specifically, for GaAs, we used data from DFT calculations as reported previously[26] to obtain the TB model. We randomly selected MD snapshots at a temperature of 400 K to calculate the electronic structure of a $2 \times 2 \times 2$ supercell of the primitive structure, comprising 64 atoms, with a $3 \times 3 \times 3$ $k$-point grid, for training and validating the ML model with 10 and 48 calculations, respectively. The DFT electronic structure calculations for comparison with HAMSTER were performed for a $4 \times 4 \times 4$ supercell of the diamond structure, comprising 128 atoms, using a $3 \times 3 \times 3$ $k$-point grid, and randomly selecting 100 MD snapshots at a temperature of 400 K.

For CsPbBr$_3$, we set the plane-wave kinetic energy cutoff to 300 eV, and sampled the electronic structure of the unit cell on a $12 \times 12 \times 12$ $k$-point grid to generate data for the TB model. We selected 24 random snapshots from an MD trajectory at a temperature of 425 K to calculate the electronic structure of a $2 \times 2 \times 2$ supercell using a $3 \times 3 \times 3$ $k$-point grid for training and validating the ML model with 12 calculations each. For the DFT band gap calculations, we selected 100 random snapshots from MD runs at each temperature and performed self-consistent calculations for all of them using a $3 \times 3 \times 3$ $k$-point grid.

For MAPbBr$_3$, we used a cut-off energy of 400 eV and sampled the electronic structure of the unit cell on an $8 \times 8 \times 8$ $k$-point grid to obtain the data for the TB model. We selected 24 random snapshots from an MD trajectory of a $2 \times 2 \times 2$ supercell at a temperature of 300 K to calculate the electronic structure, using a $2 \times 2 \times 2$ $k$-point grid, for training and validating the ML model with 12 calculations each. For the DFT band gap calculations, we selected 100 random snapshots from MD runs at a temperature of 300 K and performed self-consistent calculations for all of them using a $3 \times 3 \times 3$ $k$-point grid. The DFT electronic structure results for the $4 \times 4 \times 4$ supercell were obtained with a sampling at the $\Gamma$-point only, for a total of 100 MD snapshots, at a temperature of 300 K.

For the calculations in HAMSTER, we used the same snapshots as in the DFT calculations when comparing the two. For the $8 \times 8 \times 8$ and $16 \times 16 \times 16$ cells, we chose a sample size of 20 snapshots each, as we

found that the electronic-structure calculations showed a faster statistical convergence with respect to sample size compared to the smaller super cells.

### Molecular dynamics calculations

All DFT-based MD calculations were performed in VASP[48] using an $NVT$ ensemble. Specifically, for GaAs, we employed a $4 \times 4 \times 4$ supercell, used the $\Gamma$-point, and a time step of 8 fs to perform a DFT-MD production run containing 4000 steps after equilibration at a temperature of 400 K.

For CsPbBr$_3$, we employed a $2 \times 2 \times 2$ supercell, $3 \times 3 \times 3$ $k$-point grid, Tkatchenko-Scheffler van der Waals corrections[50], and a time step of 8 fs to perform a DFT-MD production run containing 2000 steps at each temperature.

For large-scale MD calculations of CsPbBr$_3$, we first trained three MLFFs in VASP[5,51,52], at 425 K, 525 K, and 625 K, respectively, using the Bayesian on-the-fly scheme and a time step of 6 fs for a $4 \times 4 \times 4$ supercell of the cubic structure. In order to ensure robustness and convergence of the resulting MLFFs, the training runs lasted 100 ps. During training, the default set of hyperparameters was first employed, specifying the following tags in VASP: ML_RCUT1 = 8 Å, ML_RCUT2 = 5 Å, ML_SION1 = 0.5, ML_SION2 = 0.5. Using the DFT data collected during on-the-fly training, we trained three MACE models[9], one per temperature, using 2 message passing layers, messages with 128 channels and max_L of 2, correlation order of 3, spherical harmonics up to $l = 3$, and a radial cutoff of 10 Å. The weighted mean squared error loss function was implemented with initial energy and force weights being 1 and 100, respectively, increasing to 1000 and 100 in the last 20% of the epochs, with stochastic weight averaging enabled. The MACE models were trained with a batch size of 5 for a total of 600 epochs. The validation sets were chosen to be 10% of the configurations in each data set. The reference energies for isolated atoms were obtained through spin-polarized DFT calculations. The resulting MACE force fields were used with the LAMMPS code[53], and applied to a $16 \times 16 \times 16$ supercell of the cubic structure for 5000 time steps of 0.6 fs.

For MAPbBr$_3$, we applied a classical force field from the literature[54] using the LAMMPS code[53]. We equilibrated the system, including its lattice, in successive $NVT$ and $NpT$ runs lasting approximately 22 ps in total at each temperature, before running 5 ps long production runs from which we extracted one snapshot every 50 fs. We repeated this procedure for the different supercell sizes, namely the $2 \times 2 \times 2$, $4 \times 4 \times 4$, $8 \times 8 \times 8$, and $16 \times 16 \times 16$ supercell of cubic MAPbBr$_3$.

### Contribution of spin-orbit coupling in tight binding model

One can compute the matrix elements of the SOC Hamiltonian as

$$H_{\text{soc},ij} = \gamma \langle \chi_i | \hat{L} \cdot \hat{S} | \chi_j \rangle, \tag{7}$$

where $\gamma$ is a learnable parameter that depends on the atomic species, and $\hat{L}$ and $\hat{S}$ denote the orbital and spin-angular momentum operators, respectively. In the case of a basis of AOs given in spherical harmonics, we can analytically evaluate Eq. (7) (see, e.g., ref. 55). For systems described using hybrid orbitals (HOs), as employed in our previous work on GaAs[26], the SOC Hamiltonian in the HO basis is computed via the transformation

$$H_{\text{soc}}^{\text{ho}} = \mathbf{F}^{\dagger} H_{\text{soc}} \mathbf{F}, \tag{8}$$

where the matrix, $\mathbf{F}$, contains the coefficients that expand the HOs in terms of AOs (see Tab. I in ref. 26). We note that this model only includes on-site SOC interactions; off-site contributions and spin-flip terms between different atomic centers are neglected.

## Principal component analysis

To gain deeper insight into the descriptor space, we performed a PCA on the samples selected by the ML model. Prior to PCA, each descriptor dimension is rescaled to the interval [0, 1], and a consistent basis transformation was applied across datasets to ensure comparability. For calculations with an increasing number of training structures, new structures were added incrementally to the set while retaining all previously included ones.

## Sampling of kernel support points

From the entire set of descriptor vectors for all training structures, a subset of $N_p$ kernel support points is sampled using k-means clustering with $N_{cluster}$ clusters. To account for the varying importance of each cluster, we weighted the number of points sampled from the $i$-th cluster by

$$w_i = \alpha s_i + (1 - \alpha)v_i, \tag{9}$$

where $s_i$, $v_i$ are the normalized size and variance of the $i$-th cluster, respectively, and $\alpha = 0.5$ for all our calculations. At least one point was sampled from each cluster to ensure that each type of interaction was sampled at least once.

## Parameter optimization

We perform separate training runs to optimize each set of model parameters, thereby avoiding numerical instability. Specifically, we optimize $V_\nu$ for the TB model (Eq. (3)), $h_n$ for the ML model (Eq. (4)), and the $\gamma$ parameters for the SOC model (Eq. (7)) using gradient descent with the ADAM optimizer[56]. For the TB parameters, learning rates between 0.1 and 0.3 were found to work well, and all TB parameters were initialized to 1. The ML corrections are expected to be small, so a smaller learning rate of 0.01 or 0.005 is used, with all ML parameters initialized to 0. SOC parameters are also initialized to 1. Since the SOC model has only a few parameters, it is largely insensitive to the learning rate and converges within just a few iterations.

We optimize the TB model using the non-distorted primitive unit cell and using 12 randomly selected snapshots from MD as validation data to prevent overfitting; the ML and SOC models are optimized using 12 randomly selected snapshots from an MD run at a single temperature, as described above.

The TB model is optimized for up to 1000 training steps, with early stopping to prevent overfitting. For the ML and SOC models, we first run a combined training of 200 optimization steps, during which both ML and SOC parameters are fitted simultaneously. Next, the ML parameters are reinitialized to 0 and refitted for an additional 200 steps while keeping the SOC parameters fixed. This two-stage procedure is practical because the ML and SOC models strongly influence the band gap and are interdependent.

At each iteration, the gradient of the loss function with respect to each parameter is required. Using the chain rule, this can be expressed as

$$\frac{\partial L}{\partial p} = \sum_{nkij\mathbf{R}} \frac{\partial L}{\partial E_{n\mathbf{k}}} \frac{\partial E_{n\mathbf{k}}}{\partial H_{ij}^{\mathbf{k}}} \frac{\partial H_{ij}^{\mathbf{k}}}{\partial H_{ij}^{\mathbf{R}}} \frac{\partial H_{ij}^{\mathbf{R}}}{\partial p}, \tag{10}$$

where $p$ denotes a model parameter and $\mathbf{R}$ are the lattice translation vectors.

The term on the right of Eq. (10), $\frac{\partial H_{ij}^{\mathbf{R}}}{\partial p}$, is straightforward to compute since the model parameters enter the Hamiltonian matrix elements linearly. The term, $\frac{\partial E_{n\mathbf{k}}}{\partial H_{ij}^{\mathbf{k}}}$, can be evaluated using the Hellmann-Feynman theorem and the eigenvectors obtained from the diagonalization of the Hamiltonian. Using Eq. (1), we can identify the derivative, $\frac{\partial H_{ij}^{\mathbf{k}}}{\partial H_{ij}^{\mathbf{R}}}$, simply as $e^{i\mathbf{k}\cdot\mathbf{R}}$. Finally, the first term, $\frac{\partial L}{\partial E_{n\mathbf{k}}}$, depends on the chosen loss function. In our work, we adopted a weighted MAE to

quantify the discrepancy between predicted and target energy eigenvalues:

$$L = \frac{1}{\sum_n w_n \sum_{\mathbf{k}} w_{\mathbf{k}}} \sum_{n\mathbf{k}} w_n w_{\mathbf{k}} |E_{n\mathbf{k}}^{target} - E_{n\mathbf{k}}^{pred}|, \tag{11}$$

where $w_n$, $w_{\mathbf{k}}$ are the weights of the $n$-th band and the $k$-point $\mathbf{k}$, respectively, and, $\Sigma_n w_n$, $\Sigma_{\mathbf{k}} w_{\mathbf{k}}$ are the sum of the respective weights. To further accelerate the gradient computation, which is the main bottleneck during optimization, we explicitly made use of the sparsity of the Hamiltonian to only compute elements of the sum over $i$, $j$ in Eq. (10) for which the respective matrix element $H_{ij}^{\mathbf{R}}$ is non-zero.

Nevertheless, our optimization strategy has some limitations. First, the TB, ML, and SOC parameters are trained in separate stages to avoid numerical instability, which prevents fully capturing their mutual interdependence. Consequently, subtle correlations between these components may not be entirely reflected in the final model. Second, gradient-based optimization may be less effective if the Hamiltonian exhibits strong nonlinear dependence on its parameters, and it cannot guarantee convergence to the global minimum. Despite these limitations, our approach provides a stable and computationally efficient framework for fitting effective Hamiltonians.

## Hyperparameter optimization

For parameters that cannot be efficiently optimized using gradient descent, we perform hyperparameter optimization using a Tree-structured Parzen Estimator (TPE)[57]. In each hyperparameter iteration, the TPE proposes a new candidate using a Bayesian estimator. For each candidate, a full parameter optimization is performed, with the model parameters reinitialized at the start of the run. Once the optimal hyperparameter setting is identified, a final optimization is carried out to obtain the corresponding optimal model parameters.

For the TB model, the effective core charge parameter (see Sec. C in ref. 26) is treated as a hyperparameter, since computing its gradient is impractical. For the ML model, the hyperparameters include the number of kernel support points $N_p$, the number of clusters $N_{cluster}$, and the kernel width $\sigma$. As described in the main text, the structural entries in the descriptor can either be fixed or dynamically updated based on lattice distortions. For CsPbBr$_3$ and GaAs, the structural entries are kept fixed. For MAPbBr$_3$, they are updated dynamically to account for thermal lattice changes. In this case, the dynamic entries are normalized by the cutoff radius or the maximum angle $2\pi$, and the environment defined in Eq. (5) is scaled by a fixed factor, env_scale, which is treated as an additional hyperparameter. The final ML models comprise 250 parameters for GaAs, 910 for CsPbBr$_3$, and 420 for MAPbBr$_3$.

## Band window selection and weighting

As described in detail above, the training and validation datasets consisted of energy eigenvalues obtained from DFT. Here, the choice of the energy window containing the bands used in the optimization must be made carefully, as it depends on the pseudopotential used in the DFT calculations, and the number of Kohn-Sham orbitals is generally larger than the number of basis states in the TB basis. For CsPbBr$_3$ and MAPbBr$_3$, we included the nine uppermost valence bands together with additional conduction bands to account for contributions of the remaining orbitals (five for CsPbBr$_3$, four for MAPbBr$_3$). When computing the loss, increased weight was assigned to the $\Gamma$-point and to the highest valence and lowest conduction bands, while reducing the weight of the two uppermost bands. These numbers were doubled when SOC is included in the calculation and further scaled with the size of the supercell. The reason for this is that these bands typically exhibit larger errors, which can disproportionately influence the fit as the model attempts to minimize their contributions. Moreover, they may include components from higher-lying conduction

bands due to band entanglement, making them difficult for the model to predict with sufficient accuracy. Since these states lie far from the band gap region, they are also less critical for most target applications, further justifying their reduced influence during training.

## Data availability

The source data that support the findings of this study are available in the Zenodo repository with the identifier https://doi.org/10.5281/zenodo.18485403[58].

## Code availability

The code used in this study is available at GitHub (https://github.com/TheoFEM-TUM/Hamster.jl). The calculations were performed using version 0.2.1, which is archived on Zenodo[59].

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

## Acknowledgements

We thank Kaiwen Chen and Jonas Oldenstaedt (TU Munich) for assistance with numerical aspects of the calculations. Funding provided by the Bavarian Collaborative Research Project Solar Technologies Go Hybrid (SolTech), the Deutsche Forschungsgemeinschaft via SPP2196 Priority Program (Project-ID: 424709454), Germany's Excellence Strategy-EXC 2089/1-390776260, and the Studienstiftung des Deutschen Volkes, is gratefully acknowledged. The authors further acknowledge the Gauss Center for Supercomputing e.V. for providing computing time through the John von Neumann Institute for Computing on the GCS Supercomputer JUWELS at Jülich Supercomputing Center.

## Author contributions

M.S. carried out the research with support from S.Z., F.V., and F.P.D. M.S. and D.A.E. prepared the manuscript with input from all authors. D.A.E. conceived and supervised the project.

## Funding

## Competing interests

The authors declare no competing interests.
