## [Transparent Peer Review file · Nature Communications]

Physics-informed Hamiltonian learning for large-scale optoelectronic property prediction

Corresponding Author: Professor David Egger

Version 0:

Reviewer comments:

Reviewer #1

(Remarks to the Author)

The manuscript aims to contribute to the prediction of optoelectronic properties in large-scale materials. It develops a machine learning model based on tight-binding density functional theory, with the parameters of an effective Hamiltonian fitted using a kernel regression approach. The manuscript claims improved efficiency in computing eigenvalues.

However, the text is difficult to follow for non-specialist readers. Several equations or derivations are omitted and need further discussion or proper citation. For example, Equation 4 (the descriptor x_i) is unclear, and its relationship to Equation 5 is not explained. The optimization and fitting of parameters and hyperparameters are confusing. Authors should discuss the limitations of these method.

Several figure captions are incomplete; for instance, Figure 2a does not clearly state which property is studied. Clusters in Fig 2b are not clear. Figure 3 omits values for the Hamster(16) model. Fig 4 captions has typos.

Code repository misses step on how to form the required h5 files, and how to form the configuration and main input files.

The scope of the study may be better suited to more specialized journals, such as npj Computational Materials. I do not recommend the current version for publication in Nature Communications.

(Remarks on code availability)

Code repository misses step on how to form the required h5 files, and how to form the configuration and main input files.

Reviewer #2

(Remarks to the Author)

The authors present an interesting study on data drive Δ ML learning of Hamiltonians of complex systems. The article is well motivated. I have the following queries :

1. If domain of application is optoelectronics, then would you not say modeling excited states are crucial ?.
2. Referring to "However, ENNs typically require large datasets for training, which can be a limiting factor from a computational perspective."
One major reason why equivariant NN method gained prominence was that they reduced the training sample size requirement and made neural networks more data efficient. So if this sentence is written with in a specialistic context, then it should be amended.
3. It looks like you are using Δ ML to learn the different between tight binding and ground truth of a Hamiltonian matrix element. Can you pre-train you model Hamiltonian to some specific chemical systems and fine tune on completely different chemical system ? . Kindly discuss/ provide clarity on this fundamental transferability challenges related to addressing this question.
4. The TB Hamiltonian is defined in Fourier Space to account for periodic boundary condition (equation 1). Is the machine learned part of the delta learning (Equation 4) agnostic toward real space or periodic formulation of equation 1 ? .

5. Would you not want equation (5) also to have Fourier space formulation to account for periodic effects given equation (1) (also a local descriptor will not be able to capture non local effect by the virtue of fixed cutoff) ?. And this does not look like an equivariant formulation.

6. Additional comments:

- Please give clear explanations with respect to variables in Figure 1b.
- I would also like to see Figure 1c plotted in RMSE including a curve with out the delta ML correction (i.e. pure TB) and see how much RMSE improves relatively when ML correction is added. Would you also kindly provide a similar result for MAPbBr₃ ?
- How are the residuals estimated (general comment)?.

7. Minor comment :

The quality of the plotting could be better (e.g. parts of Fig 2, 4 are not legible).

(Remarks on code availability)

There is a README section in the GitHub detailing installation instruction

Reviewer #3

(Remarks to the Author)

The manuscript “Physics-informed Hamiltonian learning for large-scale optoelectronic property prediction” by Schwade et al. presents a model that follows a “learn-the-rules, not the answers” approach. Instead of directly predicting target properties, the authors learn the parameters of a physically motivated Hamiltonian model and then compute observables from it. While such approaches are usually more challenging to design than end-to-end deep neural networks, they offer important advantages. The resulting models encode physics beyond standard rotational, translational, and permutational symmetries, are more interpretable, and typically generalize better from limited data.

The authors demonstrate the effectiveness of their HAMSTER model for GaAs, CsPbBr₃, and MAPbBr₃, achieving mean absolute errors in band-gap prediction on the order of 50 meV with only a small number of reference structures. They also show good scaling to larger system sizes, up to approximately 20k atoms, which is particularly relevant for realistic device-level simulations where electronic properties must be evaluated on large configurations.

The manuscript is clearly written, well structured, and the results are convincingly presented. I have only one minor suggestion: it would be very helpful if the authors could provide estimates of computational cost as a function of system size (e.g., wall-clock time for representative system sizes). Since the scalability of the approach is one of its key strengths, these practical performance details support readers in assessing applicability to their own problems.

With this minor remark, I strongly support publication in Nature Communications. The approach is a significant and timely contribution to physics-informed machine learning for materials modeling, and it will be of broad interest to both academic and applied research communities.

(Remarks on code availability)

Version 1:

Reviewer comments:

Reviewer #1

(Remarks to the Author)

I thank the authors for their very careful consideration and thorough responses to my comments. All of my suggestions have been appropriately addressed and incorporated into the revised manuscript and/or the Supporting Information. I agree with all of the authors' responses and have no further comments. I recommend the manuscript for publication.

(Remarks on code availability)

I thank the authors for their very careful consideration and thorough responses to my comments. All of my suggestions have been appropriately addressed and incorporated into the revised manuscript and/or the Supporting Information. I agree with all of the authors' responses and have no further comments. I recommend the manuscript for publication.

Reviewer #2

(Remarks to the Author)

My authors have addressed my queries

(Remarks on code availability)

Reviewer #3

(Remarks to the Author)

As in my previous assessment, I regard the manuscript titled "Physics-informed Hamiltonian learning for large-scale optoelectronic property prediction" as a valuable and timely contribution to the field. The work addresses a significant challenge in computational materials science: the accurate and scalable prediction of optoelectronic properties beyond the system sizes accessible to conventional ab initio methods.

I concur with the other referees on the importance of clarifying certain methodological aspects and of more explicitly discussing the limitations of the present framework in capturing excited-state phenomena. Addressing these points was important for strengthening the manuscript's clarity and impact. That said, I consider the methodological progress demonstrated in this work to be both substantial and of broad interest. The proposed physics-informed Hamiltonian learning framework represents a significant step toward overcoming the computational bottlenecks in large-scale simulations of optoelectronic properties. Given the central role of such properties across a wide range of practical applications, from photovoltaics to semiconductor device design, extending predictive modeling to realistic system sizes is utterly important.

Overall, I find the manuscript well written, clearly structured, and technically sound. In my view, it merits publication in Nature Communications.

(Remarks on code availability)

I could use the code

Reviewer 1:

Comment 1: The manuscript aims to contribute to the prediction of optoelectronic properties in large-scale materials. It develops a machine learning model based on tight-binding density functional theory, with the parameters of an effective Hamiltonian fitted using a kernel regression approach. The manuscript claims improved efficiency in computing eigenvalues.

Response: We thank the reviewer for the review of our work. We addressed the reviewers' concerns below.

Comment 2: However, the text is difficult to follow for non-specialist readers. Several equations or derivations are omitted and need further discussion or proper citation. For example, Equation 4 (the descriptor x_i) is unclear, and its relationship to Equation 5 is not explained. The optimization and fitting of parameters and hyperparameters are confusing. Authors should discuss the limitations of these method. Several figure captions are incomplete; for instance, Figure 2a does not clearly state which property is studied. Clusters in Fig 2b are not clear. Figure 3 omits values for the Hamster(16) model. Fig 4 captions has typos. Code repository misses step on how to form the required h5 files, and how to form the configuration and main input files.

The scope of the study may be better suited to more specialized journals, such as npj Computational Materials. I do not recommend the current version for publication in Nature Communications.

Response: We thank the reviewer for this valuable feedback. We have carefully revised the manuscript to improve clarity and accessibility. In particular, we have added additional explanations where equations or derivations were previously omitted, clarified the definition of the descriptor in Eq. 4 and its relationship to Eq. 5, and improved the description of the parameter optimization and hyperparameter fitting procedure. We now also include an explicit discussion of the limitations of our method.

We have further revised the parameter optimization for CsPbBr₃ after identifying a numerical instability in large supercells at high temperatures following the transition to the MACE potential from VASP's MLFF. For consistency, we have also re-evaluated all calculations for MAPbBr₃, although no analogous instability was observed for that system.

We appreciate the reviewer's suggestion regarding the appropriate scope of the study. We believe that the substantial improvements to clarity, documentation, and methodological transparency significantly strengthen the manuscript, and we hope that the revised version now meets the standards for publication in *Nature Communications*.

Changes:

- We have augmented the explaining paragraph for Eq. 4 (p. 3, left column), which now reads:
To this end, each non-zero matrix element in the Hamiltonian of Eq. 3, determined by a cutoff radius r_{cut} , is associated with a descriptor vector, \mathbf{x}_{ij}^R , whose specific construction is detailed below. From the full set of descriptor vectors, we select a subset of size N_p using the k -means clustering algorithm. These serve as kernel support points and are used to predict the corrections for remaining or unseen matrix elements (see Methods for details on the sampling procedure).
- We have added additional information on how the descriptor vectors are constructed on p. 4, left column, which reads:

The descriptor vector then reads

$$x_{ij}^{\mathbf{R}} = (Z_i, Z_j, \Delta r_{ij}^{\mathbf{R}}, \theta_i^{\mathbf{R}}, \theta_j^{\mathbf{R}}, \varphi_{ij}^{\mathbf{R}}, h_{\text{env},i}, h_{\text{env},j})^T,$$

where $\Delta r_{ij}^{\mathbf{R}}$ is the distance between atoms i and j with the translation vector \mathbf{R} applied to atom j , $\theta_x^{\mathbf{R}}$ ($x = i, j$) is the angle between the orientation of the orbital on atom x and the bonding axis; and $\varphi_{ij}^{\mathbf{R}}$ is the angle between the orbital orientations on atoms i and j .

- We rewrote and restructured the (hyper-)parameter optimization, please refer to the highlighted section in the revised manuscript on page 9-11.

- We added discussion of limitations of our approach at appropriate places:

On p. 8, left column as well as right column (see comments 2 and 4 by reviewer 2)

On p. 10, right column:

Nevertheless, our optimization strategy has some limitations. First, the TB, ML, and SOC parameters are trained in separate stages to avoid numerical instability, which prevents fully capturing their mutual interdependence. Consequently, subtle correlations between these components may not be entirely reflected in the final model. Second, gradient-based optimization may be less effective if the Hamiltonian exhibits strong nonlinear dependence on its parameters, and it cannot guarantee convergence to the global minimum. Despite these limitations, our approach provides a stable and computationally efficient framework for fitting effective Hamiltonians.

- We improved visual clarity of Fig. 2b and revised and expanded the text describing the procedure and results on page 5.

- We revised the captions of Figs. 3 and 4 and corrected typos.

- We have added data for large-scale calculations for CsPbBr₃ after switching to MACE molecular dynamics (see updated figure below). We have reevaluated all calculations to ensure numerical stability and consistency. We have added a discussion of the temperature dependence of the band gap in comparison to experimental data on p. 6, left and right column:

This improved statistical convergence allows us to assess the temperature dependence of the band gap. While the absolute value of the band gap is affected by the known inaccuracies of PBE-level DFT, the resulting temperature dependence can still be meaningfully compared to experiment. For the $16 \times 16 \times 16$ supercell, we obtain a slope of 3.95×10^{-4} eV/K, in good agreement with the experimental value of 3.41×10^{-4} eV/K reported in Ref. 40. We note, however, that thermal expansion is not included in the simulation and that the temperature range considered here extends beyond that of the experimental study, both of which are expected to affect the band gap values.

We have also added a description how the MACE models are trained to the methods section on p. 9, left column, which reads:

Using the DFT data collected during on-the-fly training, we trained three MACE models [9], one per temperature, using 2 message passing layers, messages with 128 channels and `max_L` of 2, correlation order of 3, spherical harmonics up to $l = 3$, and a radial cutoff of 10. The weighted mean squared error loss function was implemented with initial energy and force weights being 1 and 100, respectively, increasing to 1000 and 100 in the last 20% of the epochs, with stochastic weight averaging enabled. The MACE models were trained with a batch size of 5 for a total of 600 epochs. The validation sets were chosen to be 10% of the configurations in each data set. The reference energies for isolated atoms were obtained through spin-polarized DFT calculations. The resulting MACE force fields were used with the LAMMPS code [53], and applied to a $16 \times 16 \times 16$ supercell of the cubic structure for 5000 time steps of 0.6 fs.

Fig. 3 Transferability of the model across temperatures and large-scale calculations.

Comment 3: Code repository misses step on how to form the required h5 files, and how to form the configuration and main input files.

Response: We have expanded the documentation to include the requested information and also included links within the README itself. For your convenience, we provide links here: main input file, structure data, eigenvalue data

Reviewer 2:

Comment 1: The authors present an interesting study on data drive Δ ML learning of Hamiltonians of complex systems. The article is well motivated. I have the following queries :

Response: We thank the reviewer for their positive assessment of our work and constructive feedback. We gladly address the remaining comments below.

Comment 2: If domain of application is optoelectronics, then would you not say modeling excited states are crucial ?.

Response: While we agree with the reviewer that excited-state properties can be an important aspect in the modeling of optoelectronic materials, oftentimes researchers are studying merely the ground-state properties of such materials with methods like DFT. Here, our framework provides a very interesting alternative for large-scale computations. That said, in principle, our framework can also be extended to construct effective Hamiltonians for excited states, provided appropriate reference data (e.g., excited-state energy eigenvalues) are available. Likewise, the method is flexible with respect to the level of theory used in the underlying electronic-structure calculations that generate the training data.

However, extending the present study to explicitly model excited states lies beyond the scope of the current work.

Changes: We now explicitly mention excited states in the discussion on p. 8, right column which reads:

These could include hybrid functionals or many-body perturbation theory, potentially extending the framework to including excited-state effects that can play an important role in the modeling of optoelectronic properties.

Comment 3: Referring to “However, ENNs typically require large datasets for training, which

can be a limiting factor from a computational perspective.” One major reason why equivariant NN method gained prominence was that they reduced the training sample size requirement and made neural networks more data efficient. So if this sentence is written with in a specialistic context, then it should be amended.

Response: We thank the reviewer for pointing this out. We agree that a key reason for the success of equivariant neural networks is their improved data efficiency, as enforcing physical symmetries significantly reduces the amount of training data required compared to non-equivariant models.

Our original wording was misleading in a specialist context and did not sufficiently acknowledge this advantage. Our intent was to contrast ENNs with physics-based and hybrid approaches that, in some cases, can achieve comparable accuracy with even fewer training samples by incorporating stronger physical priors. We have revised the text accordingly to clarify this distinction and to better reflect the established strengths of ENNs.

Changes: The statement about data efficiency of ENNs on p. 1, right column now reads:

Although ENNs improve data efficiency by encoding symmetry and therefore require fewer samples than comparable non-equivariant models, they may still demand substantial amounts of training data to capture complex dynamical behavior, such as that exhibited in multi-component materials at finite temperature.

Comment 4: It looks like you are using Δ ML to learn the different between tight binding and ground truth of a Hamiltonian matrix element. Can you pre-train you model Hamiltonian to some specific chemical systems and fine tune on completely different chemical system ? . Kindly discuss/ provide clarity on this fundamental transferability challenges related to addressing this question.

Response: The reviewer raises an important point about chemical transferability in our Δ ML framework. In principle, transfer learning is possible when the source and target systems share similar chemical environments and orbital character. For example, one could pre-train the model on a simple perovskite, such as CsPbBr_3 , and then fine-tune it on a mixed-halide system (e.g., $\text{CsPbBr}_x\text{I}_{3-x}$). Because the chemical composition and local bonding motifs remain similar, the learned parameters should serve as a meaningful starting point for the more complex system.

It is also conceivable to initialize parameters for one halogen (e.g., Br) as a starting point for another (e.g., I), provided they have comparable valence electron configurations. However, we do not expect such transferability to hold in general when the chemical environments, orbital symmetries, or electronic properties differ substantially. In such cases, the parameters learned during pre-training may not remain valid, and the model would require extensive re-training. Thus, chemical transferability is feasible but we expect it to be inherently limited by the degree of similarity between the source and target material systems.

Changes: We have added a paragraph that discusses chemical transferability on p. 8, left column, which reads:

An important consideration that follows from this capability is chemical transferability. In principle, transfer learning is feasible when the source and target systems share similar local environments, bonding motifs, and orbital character, allowing pre-trained parameters to serve as meaningful initializations. This may hold for materials that differ only by modest compositional changes or chemically similar substitutions. However, when electronic structure or orbital symmetries differ substantially, previously learned parameters are unlikely to remain valid, and retraining becomes necessary. At the same time, these considerations highlight the longer-term

potential for developing a more general model that spans broader classes of materials and can be efficiently adapted to new systems with minimal additional training.

Comment 5: The TB Hamiltonian is defined in Fourier Space to account for periodic boundary condition (equation 1). Is the machine learned part of the delta learning (Equation 4) agnostic toward real space or periodic formulation of equation 1 ? .

Response: While the TB Hamiltonian is expressed in Fourier space to incorporate periodic boundary conditions, the underlying matrix elements are computed in real space. In our Δ ML scheme, the ML model operates directly on these real-space atomic environments. Translational symmetry is automatically respected because periodic images of atoms share identical local environments, leading to identical TB and ML terms. Consequently, the model is effectively agnostic to whether the Hamiltonian is later represented in real or reciprocal space; the periodicity is fully encoded through the real-space atomic positions that serve as input.

Changes: We have added a short discussion of the translational symmetry of the environment component of the descriptor on p. 4, left column which reads:

The local environment is identical for all lattice translation vectors, \mathbf{R} , as a consequence of translational symmetry. Note that the integrals in Eq. 5 reduce to those in Eq. 3 when $V_v = 1$, and therefore need to be evaluated only once for both the TB and ML contributions.

Comment 6: Would you not want equation (5) also to have Fourier space formulation to account for periodic effects given equation (1) (also a local descriptor will not be able to capture non local effect by the virtue of fixed cutoff) ?. And this does not look like an equivariant formulation.

Response: We agree with the reviewer that our current approach is strictly local and therefore cannot capture long-range or non-local effects. A reciprocal-space formulation could, in principle, incorporate such interactions; however, it would also destroy the sparse structure of the Hamiltonian and significantly increase computational cost. Incorporating message-passing or graph-based neural network architectures is indeed a promising future direction for addressing non-local contributions, although such models may require significantly larger training datasets.

Regarding equivariance, our descriptor and TB-based construction enforce the required symmetry constraints by design through the analytic structure of the Hamiltonian. This approach differs from typical equivariant formulations, which embed equivariance directly into the network architecture. Instead of transforming equivariantly under input changes, the Hamiltonian generated by our model is fully invariant under transformations that should leave it unchanged. We have revised the manuscript to clarify this distinction.

Changes: We revised the paragraph about equivariance which now reads
Finally, we emphasize that the ordering of values within the descriptor vector in Eq. 6 is critical, as inconsistent arrangements can violate the symmetry properties of the Hamiltonian. To preserve these symmetries, the descriptor is constructed such that its structure remains invariant under operations that should produce equal matrix elements, for example the relation $H_{ij}^R = H_{ji}^{-R}$ (see, e.g., Ref. 16 for a detailed discussion of Hamiltonian symmetry properties). This should be distinguished from equivariance, which instead ensures that the model output transforms consistently under symmetry operations, as employed in equivariant models [13-15, 18, 31].

Comment 7: Additional comments: - Please give clear explanations with respect to variables

in Figure 1b.

- I would also like to see Figure 1c plotted in RMSE including a curve with out the delta ML correction (i.e. pure TB) and see how much RMSE improves relatively when ML correction is added. Would you also kindly provide a similar result for MAPbBr₃ ?
-How are the residuals estimated (general comment)?.

Response: We gladly provide Fig. 1c plotted in RMSE (excluding the uppermost conduction bands which carry reduced weight during optimization, see below) as well as an analysis of the training and validation error for MAPbBr₃ versus the number of training structures (see below).

Residuals are computed as the difference between HAMSTER and DFT eigenvalues averaged over k-points and structures.

Changes: We have added a horizontal line in Fig. 1c to indicate the error of the TB model. For Fig. 2a, we have added a data point for zero training structures that indicates the error without ML corrections.

We revised the caption of Fig. 1b which now reads:

Schematic visualization of the environment descriptor for the matrix element between atoms i and j , which treats local environments of the two atoms separately. Different atomic species are indicated by purple and gray colors. Atoms within a distance r_{cut} , which is indicated by dashed circles, are labeled as k_x , with $x = 1, 2, 3, \dots$. The s and p orbitals of selected atoms are shown schematically as red circles and red-blue handles, respectively.

Root-mean-squared error versus number of training structures for GaAs

Training and validation mean-absolute error versus number of training structures for MAPbBr₃.

Comment 8: The quality of the plotting could be better (e.g. parts of Fig 2, 4 are not legible).

Response: We thank the reviewer for pointing out that the visual clarity of some of our figures

can be improved. We revised the figures accordingly.

Changes: We improved the presentation of Fig. 2a, Fig. 2b, Fig. 3a, Fig. 4a, and Fig. 4b.

Comment 9: There is a README section in the GitHub detailing installation instruction

Response: We thank the reviewer for assessing the availability of our code repository.

Reviewer 3:

Comment 1: The manuscript “Physics-informed Hamiltonian learning for large-scale optoelectronic property prediction” by Schwade et al. presents a model that follows a “learn-the-rules, not the answers” approach. Instead of directly predicting target properties, the authors learn the parameters of a physically motivated Hamiltonian model and then compute observables from it. While such approaches are usually more challenging to design than end-to-end deep neural networks, they offer important advantages. The resulting models encode physics beyond standard rotational, translational, and permutational symmetries, are more interpretable, and typically generalize better from limited data.

The authors demonstrate the effectiveness of their HAMSTER model for GaAs, CsPbBr₃, and MAPbBr₃, achieving mean absolute errors in band-gap prediction on the order of 50 meV with only a small number of reference structures. They also show good scaling to larger system sizes, up to approximately 20k atoms, which is particularly relevant for realistic device-level simulations where electronic properties must be evaluated on large configurations.

The manuscript is clearly written, well structured, and the results are convincingly presented. I have only one minor suggestion: it would be very helpful if the authors could provide estimates of computational cost as a function of system size (e.g., wall-clock time for representative system sizes). Since the scalability of the approach is one of its key strengths, these practical performance details support readers in assessing applicability to their own problems.

With this minor remark, I strongly support publication in Nature Communications. The approach is a significant and timely contribution to physics-informed machine learning for materials modeling, and it will be of broad interest to both academic and applied research communities.

Response: We thank the reviewer for their kind and constructive feedback and gladly included new data and further discussion about the scaling of our approach in the revised manuscript.

Changes: We have added Fig. 4c (see updated figure below) that shows the runtime of HAMSTER for full, partial, and no diagonalization of the Hamiltonian. We added 2 paragraphs on p. 7 that read:

To assess the computational performance of HAMSTER, we benchmark the total runtime for three calculation modes: (i) full diagonalization of the Hamiltonian, (ii) Lanczos-based partial diagonalization targeting six eigenvalues near the VBM, and (iii) construction and output of the Hamiltonian matrix in Eq. 1 without solving the eigenvalue problem. For each benchmark, the model is evaluated for 10 independent structures, using the same configurations as in Fig. 4, and all calculations are performed on a single compute node using two Intel Xeon Platinum 8168 CPUs. Full diagonalization of the largest system size ($16 \times 16 \times 16$) is not tractable on the chosen, rather modest hardware and is therefore omitted. The resulting runtimes are shown in Fig. 4c. Since the benchmarks include the first invocation of each function, we fit a function, $T(x) = T_0 + ax^n$, to the timings and subtract the constant time $T_0 = 44$ s, i.e., the initial Julia compilation overhead, which disproportionately affects benchmarks with short execution times.

The observed scaling behavior is consistent with theoretical expectations. The construction of the Hamiltonian exhibits nearly ideal linear scaling with system size. Partial diagonalization remains significantly faster than full diagonalization, but deviates from linear behavior, showing an effective scaling exponent of approximately 1.8. This deviation originates from the shift-invert transformation, which can reduce the sparsity of the Hamiltonian. In contrast, full diagonalization yields a scaling exponent of 2.8, close to the cubic scaling expected for conventional DFT implementations. It is worth noting that the absolute computational cost of DFT calculations is substantially higher, primarily due to the overhead associated with repeated self-consistent field iterations and the much larger underlying basis sets. Additional speedups for diagonalization could be achieved through more advanced algorithms, highly parallel implementations, or GPU-accelerated solvers.

Within our approach, the wall time for the largest system considered remains below one hour, demonstrating that HAMSTER can efficiently handle large-scale systems even on modest hardware. When only the Hamiltonian is required and eigenvalues are not computed, genuinely linear scaling can be achieved. Moreover, many important physical quantities, such as transport properties or the density of states, can be evaluated with up to linear scaling by exploiting the sparsity of the Hamiltonian [46, 47]. Combining our Hamiltonian learning framework with such linear-scaling methods therefore provides a promising route toward efficient simulations of large-scale systems.

Fig. 4 Band gap of MAPbBr₃ as a function of supercell size and temperature, with associated computational scaling.

Further changes:

- Added competing interests statement.
- Changed data availability statement. Source data is now available in the Zenodo repo <https://doi.org/10.5281/zenodo.18485403>.
- Added one sentence to acknowledgments.